# Design and Integration of a Wireless Stretchable Multimodal Sensor Network in a Composite Wing [note 1]

**DOI:** 10.3390/s20092528

**Published:** 2020-04-29

**Authors:** Xiyuan Chen, Loic Maxwell, Franklin Li, Amrita Kumar, Elliot Ransom, Tanay Topac, Sera Lee, Mohammad Faisal Haider, Sameh Dardona, Fu-Kuo Chang

**Affiliations:** 1Department of Mechanical Engineering, Stanford University, Building 530, 440 Escondido Mall, Stanford, CA 94305, USA; xiyuan@stanford.edu; 2Electrical and Computer Engineering Department, University of California, Los Angeles, Engineering IV Building, 420 Westwood Plaza, Los Angeles, CA 90095, USA; loicmaxwell17@ucla.edu; 3Formerly with Acellent Technologies Inc., 835 Stewart Dr, Sunnyvale, CA 94085, USA; 4Acellent Technologies Inc., 835 Stewart Dr, Sunnyvale, CA 94085, USA; franklin_li@acellent.com (F.L.); akumar@acellent.com (A.K.); 5Department of Aeronautics and Astronautics, Stanford University, Durand Building, 496 Lomita Mall, Stanford, CA 94305, USA; ehransom@stanford.edu (E.R.); tanaytopac@stanford.edu (T.T.); seralee@alumni.stanford.edu (S.L.); fkchang@stanford.edu (F.-K.C.); 6United Technologies Research Center, 411 Silver Lane, East Hartford, CT 06108, USA; sameh.dardona@kaust.edu.sa

**Keywords:** sensor networks, CMOS, wireless communication, RTD, PZT, strain gauge, aerospace composite wing, structural health monitoring (SHM), real-time sensing

## Abstract

This article presents the development of a stretchable sensor network with high signal-to-noise ratio and measurement accuracy for real-time distributed sensing and remote monitoring. The described sensor network was designed as an island-and-serpentine type network comprising a grid of sensor “islands” connected by interconnecting “serpentines.” A novel high-yield manufacturing process was developed to fabricate networks on recyclable 4-inch wafers at a low cost. The resulting stretched sensor network has 17 distributed and functionalized sensing nodes with low tolerance and high resolution. The sensor network includes Piezoelectric (PZT), Strain Gauge (SG), and Resistive Temperature Detector (RTD) sensors. The design and development of a flexible frame with signal conditioning, data acquisition, and wireless data transmission electronics for the stretchable sensor network are also presented. The primary purpose of the frame subsystem is to convert sensor signals into meaningful data, which are displayed in real-time for an end-user to view and analyze. The challenges and demonstrated successes in developing this new system are demonstrated, including (a) developing separate signal conditioning circuitry and components for all three sensor types (b) enabling simultaneous sampling for PZT sensors for impact detection and (c) configuration of firmware/software for correct system operation. The network was expanded with an in-house developed automated stretch machine to expand it to cover the desired area. The released and stretched network was laminated into an aerospace composite wing with edge-mount electronics for signal conditioning, processing, power, and wireless communication.

## 1. Introduction

The future intelligent aerospace structures will be able to “feel”, “think”, and “react” in real-time, enabling high-resolution state-sensing, awareness, self-diagnostic, and structural health monitoring capabilities [1]. A bio-inspired multifunctional structure can provide these capabilities, allowing composite structures to carry mechanical loads, sense their environment, and diagnose their health condition in real-time. The stretchable networks serve as the “nervous system” analog; in order to enable this multifunctional performance, this network must be integrated with the composite structure. The concept of distributed sensors and sensing systems is widely recognized as the most promising technology in the field of state awareness and structural health monitoring (SHM) [1,2,3,4,5,6,7,8,9,10,11]. Despite the maturity of complementary metal-oxide-semiconductor (CMOS) and microelectromechanical system (MEMS) processes, the fabrication of a microscale sensor network capable of deployment on a macroscale structure remains a challenging technical problem [12,13,14,15,16,17,18,19]. The process of deployment from microscale fabrication to meter-level implementation exposes sensitive and fragile hardware components to risk. The deployed network must also be integrated with flexible data acquisition hardware, allowing the entire system to conform to the target structure. The goal of this research is to develop not only the “nervous system” hardware but to successfully integrate this hardware with a composite structure. 

Previous work at the Structures and Composites Lab at Stanford University has laid the foundation for many fabrication and integration processes. Guo et al. [9,11] developed a new spin-coating fabrication process to enable high-resolution features in stretchable networks with temperature and piezoelectric sensors. These networks were fabricated and tested to demonstrate their functionality in advanced composite materials for aerospace applications [9,10].

Most recently, Chen et al. [18,19] added micro-patterned strain gauges to the sensor network platform and characterized the performance of the strain gauge network. Even though the network has demonstrated the capability of measuring distributed strain, vibration, and temperature, the signal to noise ratios (SNR) of the overall system were still insufficient to compete against the commercially available off-the-shelf (COTS) products. Due to the fact that the serpentine interconnects are thin and long, they inherently possess high resistances which are not ideal for resistive based sensors such as resistance temperature detectors (RTDs) and strain gauges (SGs). The micro-patterning process at that stage was not yet high-yield or economical. 

Parallel technologies have also received increased interest. Huang et al. [20] developed a two-dimensional network processed in a CMOS foundry and stretched to the desired size. Kim et al. [21] fabricated a highly stretchable and sensitive multi-dimensional strain sensor using prestrained silver nanowire percolation networks, which is advantageous over conventional single-axis strain sensor of detecting real-time “skin-like” multidimensional strain loadings. This sensor network design relies on two prestrained percolation layers intersecting with each other and can detect the x and y axes of the surface strain independently. For next-generation personalized healthcare system applications, a seamless hybridization of stretchable on-skin sensors and rigid silicon readout circuits was developed by Niu et al. [22]. Niu et al. developed a body area sensor network comprising a collection of networked sensors that can be used to monitor human physiological signals.

Lichtenwalner et al. [23] described a novel sensor design concept for flexible arrays of resistive temperature and strain sensors fabricated on polyimide sheets, such that both temperature and strain values can be determined from resistance values. A mechanical structure based on PDMS (Polydimethylsiloxane) and a mesh of multiple copper electrode strips were developed by Lee et al. [24]. By reconfiguring the connection of electrodes, the sensor is capable of tactile and proximity sensing. A technology for stretchable circuits using standard printed circuit board (PCB) fabrication and assembly methods using polymer molding techniques has also been explored [25]. Stretchability is the main feature of such a printed circuit board.

Human activity monitoring and medical device manufacture have also been active areas in the development of multifunctional, stretchable networks [26]. Kang et al. [27] developed an ultrasensitive mechanical crack-based multifunctional sensor inspired by a spider sensory system, where crack-shaped slit organs are used to detect vibrations in a structure. 

It can be inferred from the literature survey that the development of a stretchable sensor network with high signal-to-noise ratio and measurement accuracy for real-time distributed sensing and remote monitoring is still under active research. A fabrication and integration process of the sensor network must be developed for a functional prototype demonstrating the impact of the hardware of the sensor network, automated network stretch tool, flexible electronics, firmware, and software with wireless communication capabilities.

### Problem Statement

In this paper, the overarching objective is to develop an ultra-thin stretchable sensor network for multi-modal wireless sensing integrable within a prototyped composite unmanned aerial vehicle (UAV) wing. The technical challenges associated with this problem are manifold. The network must first be fabricated at the microscale, then stretched to macroscale in an automated way, finally integrated with the structure using a flexible PCB and data acquisition device.

Figure 1 shows an overview of the entire project. The sensor nodes are populated with co-fabricated resistive based sensors such as RTDs and SGs, as well as post-fabrication surface-mounted piezoelectric transducers (PZTs). A fully automated stretch tool was designed and deployed to safely and automatically stretch the network from millimeter to meter scale. The stretched network was integrated with a flexible frame with edge-mount electronics for signal conditioning, processing, power, and wireless communication capabilities. To further demonstrate the feasibility and functionality of a real-time remote sensing system, a small composite UAV wing was created, and the sensor network was laminated to this wing for impact, strain, and temperature sensing.

## 2. Method of Approach

Our strategy in reaching our objective was divided into four main subproblems, each associated with the development of a subsystem: (a) stretchable sensor network design and fabrication; (b) flexible electronics and software development; (c) sensor network deployment and integration; (d) functionality tests.

The first subproblem includes the development of network design, material selection, and fabrication processes. In solving this problem, we delivered a microfabricated stretchable sensor network designed with predefined sensor types, locations, and final stretched dimensions.

The second subproblem involves component selection, circuit design, onboard firmware development, and wireless communication. The intention of this milestone was to deliver a functional hardware and software system to acquire and process signals from the network, bridging the gap between the multimodal sensors and meaningful signals.

The third subproblem concerns the network stretch tool design and deployment as well as sensor network lamination on a composite wing. 

The final subproblem of this milestone is an integrated sensing system with real-time distributed sensing and remote monitoring functions.

After the successful integration of the system, the network was tested and validated by acquiring temperature, strain, and impact data.

## 3. Stretchable Sensor Network Design and Fabrication

### 3.1. Network Design

The stretchable sensor network designed in this study is an island-and-serpentine type network comprising sensor “islands” connected by interconnecting “serpentines” in a grid formation [28,29,30]. The serpentines are capable of unfolding, permitting the sensor network to be expanded from a 4” wafer to a rectangular active area of up to 1 m on one side. Before stretching, each serpentine has a sacrificial bridge (as shown in Figure 2a) that is designed to snap under tension, allowing the serpentine to unfold after tension is applied.

Figure 2b and Figure 3 show the sensor network design, which consists of 3 SGs (SG1, SG2, SG3), 4 RTDs (RTD1, RTD2, RTD3, RTD4), and 10 PZTs (PZT1, PZT2, …, PZT10) in a 5 × 11 nodal network. These nodes are either populated by sensors or routing pads for transmitting signals. The design was devised to fit the aspect ratio of the composite wing surface, requiring different factors of expansion in each direction. The 3 SGs are intentionally aligned along with the spar of the wing structure, while the 4 RTDs are placed near the four edges of the wing structure to capture its temperature profile. The 10 PZTs are distributed over the upper cover of the wing with a higher density near the leading edge and the center rib to detect surface impacts and structural vibration. The sensor network is designed to occupy a 384 mm × 142 mm area on the carbon fiber skin of the wing after network expansion. The dimensions of the network and the distribution of the sensors are presented in Figure 3. Before stretch, the network design dimensions were determined to be 72672 μm × 38640 μm, and each serpentine had a wire width of 98 μm (Figure 2). The network was expanded to an area of 375 mm × 130 mm, requiring elongation ratios of 5.16 and 3.36 in the horizontal and vertical directions, respectively. Figure 3 shows the dimensions of the stretched network with integrated electronics.

### 3.2. Material Selection

To increase the SNR, thus improving the measurement accuracy of the sensing system, efforts were made to reduce the Johnson–Nyquist (or “thermal”) noise. The root mean square voltage value of the thermal noise can be expressed as:(1)$$V_{J}=\sqrt{4kTBR}\;\left(Volts\right)$$
where *k* is Boltzmann’s constant (1.38 × 10^−23^ J/K), *T* is the temperature (K), *B* is the bandwidth of the system (Hz); ($B=f_{2}-f_{1}$), and *R* is the resistance (Ω).

The resistance of any conductor can be expressed as:(2)$$R=\frac{\rho L}{A}$$
where ρ is the resistivity of the material, L is the length, and A is the cross-sectional area. To reduce the thermal noise according to this expression, reducing the resistance is the most straightforward method; thus three options are available: (a) shorten the length of the conductor, (b) increase the cross-sectional area of the conductor, (c) use a conductor with lower resistivity. However, in this study, the length and width of the interconnect are predetermined by the stretchability of the sensor network. Therefore, increasing the thickness of the structure and choosing materials with higher conductivity are the only avenues for noise reduction. In light of this, aluminum (Al) was chosen to be the material for the stretchable interconnects by virtue of its low resistivity, ease of deposition (melting point 660 °C), and excellent adhesion to dielectrics. 

The effect of network resistance reduction on sensor performance was simulated using the electronic circuit emulator LTspice. A consistent, accurate voltage divider was implemented in the simulation for the RTDs. The result of RTD responses in the network simulation is shown in Table 1. The network effect causes a temperature offset in the readings that will require calibration to remove. As the interconnect resistance decreases, the temperature offset value of the RTD decreases. Accurate measurements can only be obtained when the resistance ratio of the interconnect to the RTD is around 1% [19]. The result of SG responses in the network simulation is also shown in Table 1. A Wheatstone bridge configuration with differential low-pass filters was devised for acquiring the simulated strain data. In principle, as the resistance in-network decreases, the difference in resolution between the strain gauges decreases. At the theoretical resistance value (70 Ω), the resolution differences are negligible.

### 3.3. Fabrication Method

To produce the sensor network in batches with high yield and efficiency, a recyclable microfabrication process flow was developed, as shown in Figure 4. A 4-inch silicon wafer is used as the carrier to fit for the majority of the equipment in the Stanford Nanofabrication Facility. However, the developed process is not limited to the 4-inch silicon wafer format and is adaptable to any wafer size. Moreover, since the wafer only serves as a carrier in the process, it can be reused after the sensor network is detached, reducing the cost associated with silicon wafers. Details about the microfabrication recipe can be accessed via our previous publications [19]. Here, for simplicity, only the major modifications are highlighted as follows.

Firstly, a 50/1000/50 nm Ti/Al/Ti sandwich structure was adopted to replace the gold thin film with a thickness of 200 nm for 12,350 times cost reduction of the interconnect metallic material and 13.46% cost reduction of the overall raw materials compared with our previously reported process [19]. The resistance values of the resulting horizontal and vertical interconnects are reduced by 4.67 times and averaged at 51.47 Ohm and 36.42 Ohm, respectively. Additionally, increasing the thickness of the interconnects also improves the overall robustness of the signal layer, thus suppressing the risk of open circuits.

Secondly, instead of removing the etch mask by wet etching after releasing the network, which typically causes stiction issues and ruins the sensor network, the Al layer was deliberately retained to mitigate the electromagnetic interference with radio frequencies higher than 6.67 GHz owing to the skin effect. More shielding simulation results can be found in the conference paper [18]. This extra ground layer also provides a preliminary electrical protection for preventing damage to circuits by abnormal conditions, such as overcurrent, high or low voltage. Chemically, the Al wet etchant contains corrosive and toxic acids such as HF, HNO_3_, and H_3_PO_4_. Avoiding any wet etch process eliminates the potential hazard and safety risk. Moreover, it has been reported that when the aspect ratio of the wire width to its height is smaller than 1, the stretch-induced out-of-plane deformation could be circumvented [3,11]. Therefore, thickening the sensor network profile while maintaining its pattern reduces the chance of bending or buckling failures during the network embedment.

Thirdly, etch-facilitating holes were introduced to the bonding and routing pads to effectively resolve releasing-related issues. The XeF_2_ dry etching of a sacrificial germanium layer was chosen as the release method for its high speed, clean, stiction free process. However, this dry etch process is not efficient for releasing large features due to the short penetration depth of XeF_2_. By adding etch-facilitating holes (20 µm in diameter, 100 µm in the distance) as a benchmark, it can be seen from Figure 5 that these node-wire features begin to release after 10 min, are mostly released after 30 min, and are completely released after 40 min. By contrast with the previous process that requires more than 200 min in total, the additional release holes drastically reduce the dry etching time and the chances of having residual unreleased structures.

## 4. Flexible Electronics and Software Development

The fabricated network must be carefully integrated within the target structure to produce the desired sensing effect. In order to do this, flexible PCB hardware must route signals into a data acquisition unit. The signals must also be conditioned and processed to provide useful data.

Each sensor has different network effects that affect its measurements, which must be considered in post-processing. The network system must also be capable of wireless transmission. This allows the sensor network and data acquisition electronics to be mounted in isolation, enabling an end-user to receive real-time data without the need for a direct hardware interface. 

A two-part system is required to solve this problem. The first part of this system is a flexible PCB frame, which contains the signal conditioning and data acquisition electronics for each of the sensor types. This frame directly connects to the pads surrounding the edge of the sensor network. It also must support low-power wireless transmission so that it may transmit all sensor information off-board for processing and display. The signal conditioning for each sensor type dictates the required data acquisition elements. The on-board the system controller and data transmission electronics must also be defined at this time. Once the system schematics are complete, the flexible PCB layout must be completed to match the sensor network geometry. 

The second part of the system is on the user end where the data is received, processed, saved, and then plotted on a graphical user interface (GUI). This portion begins with the definition of the specific wireless interface, which must be supported by the on-board system. A transceiver on the user end is then selected whose application programming interface (API) will be utilized in the implementation of the GUI backend. This backend will constantly be listening and waiting for data, which gets processed and sent to the user front-end upon arrival. These two portions can be developed in parallel, as long as their interface is very clearly defined.

### 4.1. Component Selection and Circuit Design

There are three fundamental subsystems in the hardware running on the flexible frame: signal conditioning, data acquisition, and data transmission. Each of these must be carefully selected to fit the sensors in the sensor network and provide the voltage range for the data to be collected. This is meant to be a low-power system, so all modules were selected to run off a single 3.3 V source. The LM3480 (Texas Instruments, Dallas, TX, USA) was selected for this, a linear regulator that steps down a 5–12 V input to the desired voltage. This allows for a variety of external power source options while also creating a clean output signal to be used by the rest of the system.

The Analog-to-Digital Converter (ADC) selected to collect information from the strain and temperature sensors was the AD7124-8 from Analog Devices (Norwood, MA, USA). It has a built-in programmable gain amplifier (PGA) to allow for small deviation measurements of the strain sensors. It is a sigma-delta 24-bit ADC, ensuring high-resolution data output for both sensor types. It can support 16 single-ended or eight double-ended channels, which works well for the four single-ended temperature and three double-ended strain measurements. The sampling rate for this ADC maxes out at 19 kS/s across all channels, meaning a different device must be used to sample the higher-frequency PZT signals. The expected signals for these sensors oscillate at 1–15 kHz, meaning a sampling frequency of at least 30 kS/s per sensor is required to digitize these signals and avoid any aliasing properly. The Atmel SAM R21G18A, an ARM Cortex-M0+ based microcontroller (Microchip Technology Inc., Chandler, AZ, USA), was selected as the main processor for this system. It provides the necessary peripherals to operate the other sub-systems, including an internal ADC with a max 350 kS/s, SPI (Serial Peripheral Interface) capabilities to interface with the AD7124-8, and internal AT86RF module for wireless communication with an off-board radio. This chip acts as a system controller while also providing a fast-enough ADC to sample the PZT sensors. In this design, to accommodate for the 10 PZT, 4 RTD, and 3 SG sensors, two of these micro-controllers were included. One will be handling the PZT data acquisition for 8 PZTs, while the other will be handling 2 PZTs and controlling the serial communication with the AD7124-8 external ADC. A block diagram of the system and the various interfaces is depicted in Figure 6.

### 4.2. Onboard Firmware Development

The firmware and software primarily run on the SAMR21 micro-controller. A single program runs on this chip, which takes care of configuring the various peripherals (ADC, timers, wireless module, etc.). It then proceeds to run an infinite loop which takes care of data acquisition and transmission according to a trigger-based interrupt system. Hardware timers were used to trigger strain and temperature measurements, ensuring that a prescribed time elapses between subsequent sampling instances. The trigger for PZT measurements is voltage-based, raising a flag when a threshold voltage is passed. A PZT voltage spike corresponds to a detected stress wave that follows an impact, so following this first trigger, data is collected and transmitted off-board for the impact localization algorithm to utilize.

One of the key challenges in programming a single-core bare-metal system such as this is the simulation of simultaneous data acquisition. To maximize the accuracy of impact localization and quantification using PZT sensors, the data from all sensors must be as time-synchronized as possible. The processing algorithm looks at the amplitude response of the PZT data for each of the individual sensors. Strain and temperature sensors are similar in this fashion because an end-user will want to track changes occurring on all 7 of these sensors simultaneously. Neither the internal micro-controller ADC nor the external ADC supports true simultaneous data acquisition. Consequently, this turns into a programming optimization problem, where the data acquisition can be serialized in the quickest way possible to simulate parallel data acquisition. The Atmel Studio IDE (Microchip Technology Inc., Chandler, AZ, USA) [31] was used for all firmware development, and while it provides some API to facilitate the firmware configuration and device programming, these functions occasionally introduce some extra, unnecessary overhead. For PZT sampling, these were broken apart and stripped down to an optimized form while still maintaining functionality and robustness. The external ADC programming was a bit different. This was accomplished through SPI, using the micro-controller as the parent node. It issues out commands to set register values on the ADC, such as sampling speed, filtering, and channel configuration. The ADC, in turn, will transmit acknowledgment messages and data back to the micro-controller. To synchronize the data as much as possible, the ADC was configured in a continuous sampling mode. This configuration maximizes the net data acquisition speed for a group of 7 points (representing the three strain and four temperature sensors), which also minimizes the latency between the first and last collected data points. This results in the closest possible synchronization. Whenever data is desired from the master (micro-controller), it drives their common clock to communicate the information back for processing and transmission. The ADC sampling rate is set at 4 kS/s. This is as high as the micro-controller can allow since it must successfully detect when a conversion is complete, and clock in the data before any subsequent conversion on the ADC completes and erases the old data.

### 4.3. Wireless Communication

IEEE 802.15.4 (Zigbee) was selected as the wireless transfer protocol for the transmission of sensor data to the off-board radios. Two main advantages of Zigbee are its ease of implementation and low power consumption. This allows the system to be easily battery-powered, widening its potential application [32]. The trade-off for low power consumption is a reduction in transmission range and data throughput, both of which do not limit the performance of the system. The data rates can reach 250 kb/s in the 2.4 GHz band in use with a reliable range of about 10–20 m. This is acceptable for this system, since the latency between the data collection and display is still minimal, in relative human terms. Two Xbee Series 1 1mW antenna and their serial adapters (Digi International, Hopkins, MN, USA) were selected as the receiver modules for the transmitted sensor data. The firmware running on these serial adapters allows the data to be serially transmitted back to the host computer, themselves being visible as serial communication ports. The GUI displays the live strain and temperature information for each sensor simultaneously, while also bringing up PZT impact locations whenever they are detected. The back end of this GUI is a multi-threaded listening program that captures data whenever it has been received on the radios. It then processes it, saves it, then sends it to the GUI thread to be displayed. A block diagram is shown in Figure 7.

## 5. Sensor Network Deployment and Integration

### 5.1. Sensor Stretch Tool Design and Deployment

After being fabricated, the network must be stretched to macroscale. Island-and-serpentine networks like the one described in this paper have been studied but have generally been expanded manually with hand tools or improvised fixtures and adhesives [33,34]. Networks of this type have several structural properties that make them difficult to manipulate by hand. Specifically, the interconnects have very low bending rigidity, allowing them to become easily displaced and tangled during handling. Tangling is effectively impossible to reverse, given the low structural strength of the interconnects, making it an existential risk to the network. Additionally, interconnects are easily snapped accidentally by human technicians. To reduce the possibility of human error, a tool was designed to stretch island-and-serpentine networks in a 2-D plane in an automated way.

During the stretching process, the sensor nodes at the edges of the network must always be constrained to be uniformly spaced, allowing them to successfully interface with the flexible PCB and associated electronics. In order to enforce this constraint, four deployable scissor-hinge structures were designed. Each unit of the scissor-hinge structure is built from two straight, equal lengths and a 1M screw acting as a central pivot. Each encapsulating node contains a 1.1 mm diameter through hole, allowing it to accommodate a 1M screw. In this way, the encapsulating nodes can be fastened to the scissor-hinge structures. Scissor-hinges can be swapped out of the assembly depending on the required number of encapsulating nodes, allowing the process to be performed for rectangular networks at a variety of aspect ratios.

The scissor-hinge mechanisms are attached to 3D-printed corner assemblies, which also contain mounts for linear rails. Inverted ball transfer mounts are press-fit into the base of the corner assemblies allowing them to traverse the build area during the stretching process. When closed, the entire assembly accommodates an unstretched sensor network with a 100 mm^2^ footprint.

To deploy the scissor-hinge assembly precisely, two sets of two synchronous belt drives were employed: one pair controlling the expansion in the x-direction, and the other controlling expansion in the y-direction (Figure 8a). Each pair of drives was driven by a pair of NEMA 17 stepper motors, each tied to a single A4998 stepper motor and Arduino Uno controller. An optical encoder allows each pair of drives to be controlled independently, allowing deployment rates between 2 mm/sec and 20 mm/sec. The deployment process occurs over a period of approximately 1 minute. Figure 8b shows a sensor network stretched over a large area using the network stretch tool.

The released sensor network (Figure 9) was handled with the network stretch tool to automatically expand it to cover an area of 375 mm by 130 mm. As shown in Figure 10, the released and stretched network is laminated into a flexible frame with edge-mount electronics for signal conditioning, processing, power, and wireless communication capabilities.

### 5.2. Sensor Network Lamination on a Composite Wing

To further demonstrate the feasibility and functionality of a real-time remote sensing system, a small composite UAV wing was created, and the integrated sensor network was laminated into the wing. Firstly, the stretched network was carefully connected to the flexible frame by applying silver paint (Ted Pella, Leitsilber 200, silver content 45%) for electrical conduction. Ten PZTs (piezo plate, 2 × 2 × 0.2 mm, APC) were then surface mounted to their designated locations within the network using the same silver paint. The etch-facilitating holes helped the silver paint permeate and anchor the perforated bond pads to avoid slip due to shear stress. Furthermore, since the silver paste can contact the conductive side of the bond pads through the perforations, the insulating side of the bond pads faces the host structure to avoid any short circuit. The connected network was installed onto the carbon fiber composite wing with a 25 cm chord length and a 50 cm wingspan. A wet lay-up consisting of mixed epoxy (L285 MGS Epoxy Resins, H287 MGS Hardener, Aircraft Spruce and Specialty Co., Corona, CA, USA), fiberglass veil, sensor network, perforated release film, bleeder-breather, and bagging film was manufactured to create a high-quality laminate. After initial curing at room temperature for one day, the laminate was vacuum cured for 15 h at 80 °C to produce the desired mechanical and thermal properties. Laminating resin MGS L285 was selected for its compatibility with aerospace applications and long pot life (approx. 4 h). It is worth noting that only the sensor network and the flexible frame were vacuumed in the vacuum-sealed bag. All the circuit components and connectors were placed outside the vacuum bag and were carefully protected from being damaged by the suction force and excess epoxy. After the vacuum bagging lamination, a close and conformal coating was created to completely cover the contour of the sensor network. The thickness of the coating (150 μm) is greatly larger than the thickness of the sensor network (30 μm), hence its appearance on the upper surface of the airfoil will not cause any disturbance to the aerodynamic performance of the composite wing. As a proof of the practical implementation of the sensor network for long term aerospace applications, previous work based on the same network integration approach has demonstrated its reliability through a wind-tunnel experiment and a fatigue test [19,35]. Finally, a commercial strain gauge (KFH-6-350-C1-11L3M3R, OMEGA Engineering, Norwalk, CT, USA) was placed close to the root of the composite wing near the network strain gauge for calibration and comparison. Figure 11 shows the stretched sensor network on a model aircraft wing with integrated electronic hardware and wireless communication capabilities. The end-to-end system, composed of data acquisition, transfer, and display, is fully functional.

## 6. Functionality Tests

After integration, a series of tests were performed to validate the sensor network system. To characterize the functionality of the sensors, the resistances of SGs and RTDs and the capacitances of PZTs were measured. The resistances of the three SGs were found to be 15.95 ± 5.03% kOhm, while the resistance of the interconnects in series with the SGs was averaged at 1.68% of the SG’s resistance. The resistances of the four RTDs were found to be 15.71 ± 3.44% kOhm, while the resistance of the interconnects in series with the RTDs was averaged at 1.23% of the RTD’s resistance. The average capacitance of the PZTs placed on the sensor network was measured as 0.45 nF. The calibration of the SGs and RTDs were based on the characterization results of our previous work [9]. The results of the validation tests are summarized in the following sections.

### 6.1. Strain Gauge Result

To verify the measurement accuracy of the SGs on the sensor network, a commercial strain gauge was placed close to the location of the SG near the root of the wing. A cantilever beam experiment was performed on the wing by clamping the root of the wing and slowly adding weights (0.5 kg) to the tip one after another. The strain data from the commercial strain gauge were collected with a National Instruments (Austin, TX, USA) NI-9236 C Series Strain/Bridge Input Module. The strain data from the network strain gauge were transmitted through Xbee Series 1 and recorded with a remote computer. The static test result is expressed in two forms. The raw data are plotted in Figure 12 to compare the measurement results from the network strain gauge with the commercial strain gauge. The experiment was conducted in a stepwise approach. Therefore, the plot is divided into several intervals, with each interval representing one specific loading condition of the wing structure. 

From the plot, we can clearly see that the noise of the commercial strain gauge is significantly higher than the network strain gauge. Quantitatively, this observation can also be verified by quantifying the fluctuation of the signals in their standard deviations, where roughly a ten-time difference can be found between the direct measurement results of commercial strain gauge and the network strain gauge. The noise floor determines the minimum detectable signal (MDS) of the measurement system. Therefore, the MDS of the commercial strain gauge was calculated as 1.9999 micro-strain compared to 0.2029 micro-strain MDS for the network strain gauges. 

Table 2 shows the comparison of the average strain values measured by the network strain gauge and the commercial strain gauge under the same conditions. Numerically, it can be further confirmed that the network strain gauge is functioning accurately. In addition to the thermal noise reduction efforts described in Section 3.2, the main reason for this improved performance is the common-mode rejection feature of the operational amplifier in our customized circuit, which can successfully suppress the noise level from sources such as power supply ripple injection and electromagnetic interference coupling.

### 6.2. Piezoelectric Transducer Result

To validate the functionality of the PZT sensors, a small impact was created by tapping on the surface of the wing. Except for PZT7 which is short-circuited due to human error during the manual mounting process, all the other nine PZT sensors responded to the stimulus in real time. Among these nine functioning PZT sensors, five of them located at the corners (PZT1, PZT2, PZT9, PZT10) and the center (PZT5) were selected to test their sensitivities. From the time-domain signal of the PZT sensors, it can be observed that the sudden spikes are caused due to the impact on the wing surface (Figure 13). When the impact is happening near a certain PZT, the magnitude of the voltage becomes larger. It is worth noting that the impact action is not perfectly controlled; thus, the responses of the PZTs are not identical. As can be seen from Figure 13, PZT10 at the bottom right corner has the highest peak voltage compared with other PZTs. In contrast, even though PZT5 was tapped, no significant signal changes can be observed throughout its time line, whereas small spikes can still be discovered from other PZT responses at around 17 s on the time axis, meaning that this PZT at the center location is not responsive to small impacts. In light of this, we analyzed the power spectral density of PZT5 in the frequency domain and found out its noise level is above others.

### 6.3. Resistance Temperature Detector Result

The RTD responses were validated by using a heat gun to increase the surface temperature of the composite to a known value, verified with a laser infrared thermometer (FLIR TG165). During which, the RTD response was saved and converted to the temperature data, as shown in Figure 14. The heat was swept on the wing from top left to bottom left to top right to bottom right. The heat wave was precisely presented by the four plots as the sequential rise of the temperature at four different locations is clearly visible. The temperature response of the four RTDs proves the functionality of the sensor network as a temperature sensor. As part of the stretchable sensor network embedded within the composite wing, the RTD sensors are all sandwiched between two fiberglass layers, which creates a thick thermal barrier blocking the heat dissipation, thus the thermal effect is transmitted tardily. In general, the temperature raises up in 5 s, whereas the cooling down process takes a much longer time, usually in the order of several minutes. It is worth noting that the RTD sensor is not limited to detecting small temperature changes (from 25°C to 35 °C) as demonstrated in this example but has been proved to be fully functional up to 70 °C [19].

## 7. Conclusions

This article presented the development of a stretchable sensor network with high signal-to-noise ratio and measurement accuracy for real-time distributed sensing and remote monitoring. A recyclable microfabrication process flow was developed to improve the yield and efficiency of the sensor network production. The sensor network consists of 3 SGs, 4 RTDs, and 10 PZTs in a 5 × 11 nodal network. The sensor network was successfully stretched with an in-house developed network stretch tool to expand it to cover a large area automatically. The design and development of a flexible frame with signal conditioning, data acquisition, and wireless data transmission electronics for a stretchable sensor network were presented. The stretchable sensors comprising electronic circuits can be integrated on unconventional surfaces for the practical applications. The current research demonstrated the design, fabrication, and integration of sensor networks on a composite UAV wing. The future path forward would be to use the sensor network for real-time state sensing, awareness, and SHM capabilities-based aerospace applications.

## Figures and Tables

**Figure 1 sensors-20-02528-f001:**
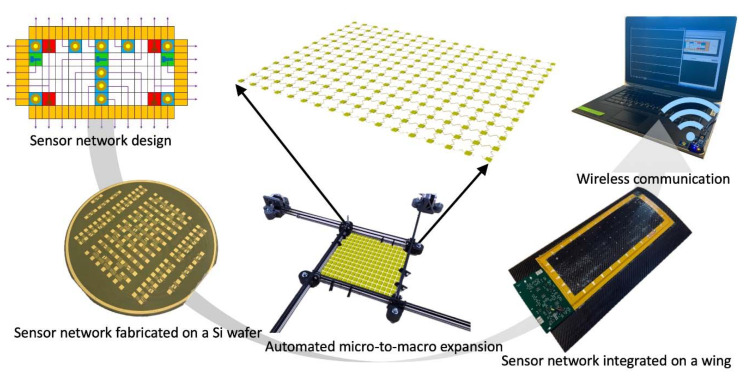
Overview of sensor network design, fabrication, stretching, integration, and wireless communication.

**Figure 2 sensors-20-02528-f002:**
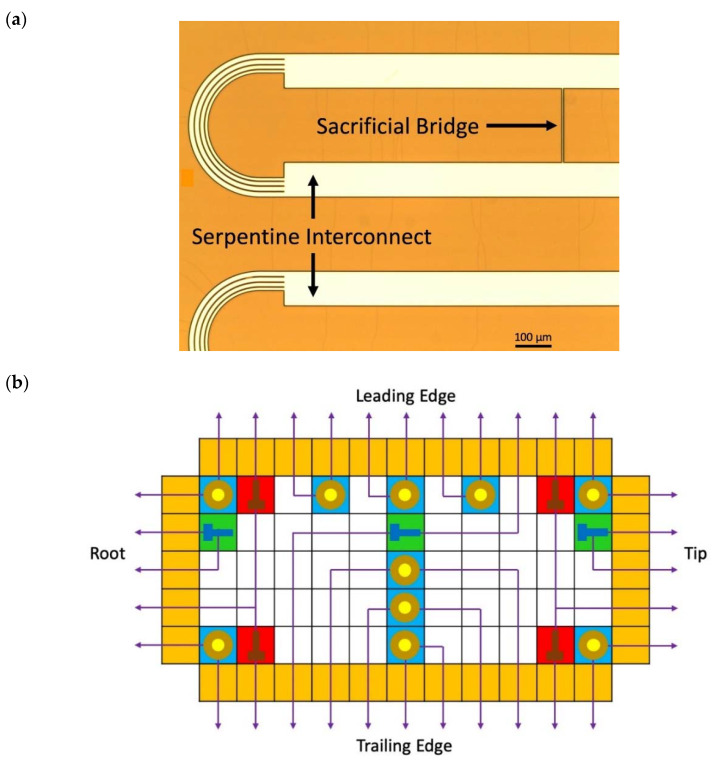
(**a**) Microscopic image of the serpentine interconnects and the sacrificial bridge; (**b**) the sensor network design consisting of 3 Strain Gauges (SGs), 4 Resistive Temperature Detectors (RTDs), and 10 Piezoelectrics (PZTs) in a 5 × 11 nodal network.

**Figure 3 sensors-20-02528-f003:**
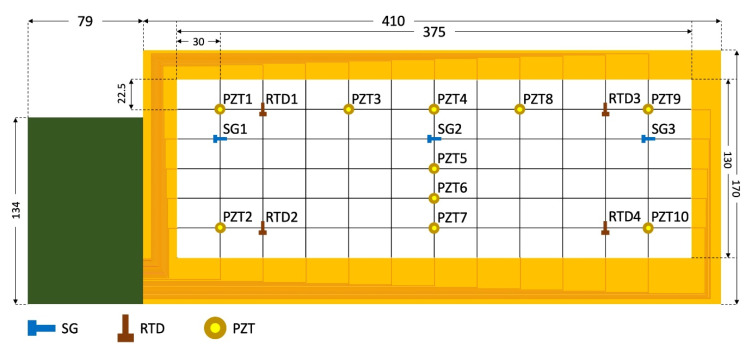
The dimensions of the network and the distribution of the sensors (all dimensions are in mm).

**Figure 4 sensors-20-02528-f004:**
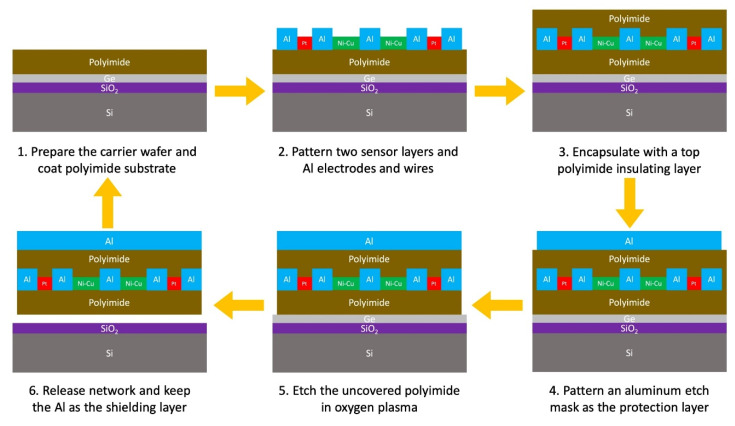
Microfabrication process flow chart.

**Figure 5 sensors-20-02528-f005:**
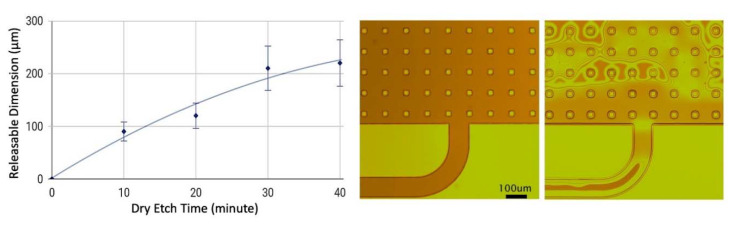
Dry etching time with released dimension (**left**) and the microscopic images (**right**) of a node-wire feature before and after 10 min of XeF_2_ dry etching.

**Figure 6 sensors-20-02528-f006:**
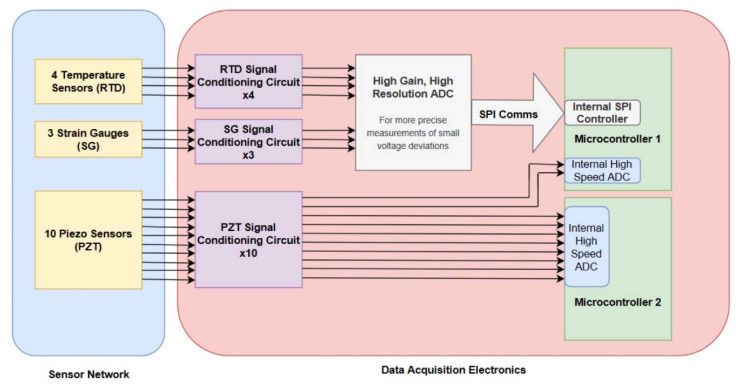
Block Diagram of Sensor-Electronics interface.

**Figure 7 sensors-20-02528-f007:**
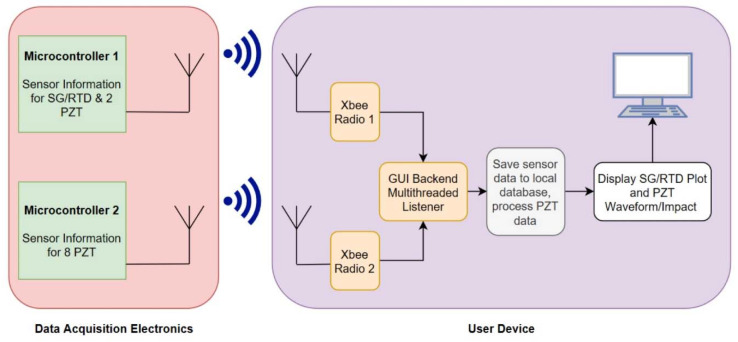
Interface and data flow between data acquisition electronics and graphical user interface (GUI) on the host computer.

**Figure 8 sensors-20-02528-f008:**
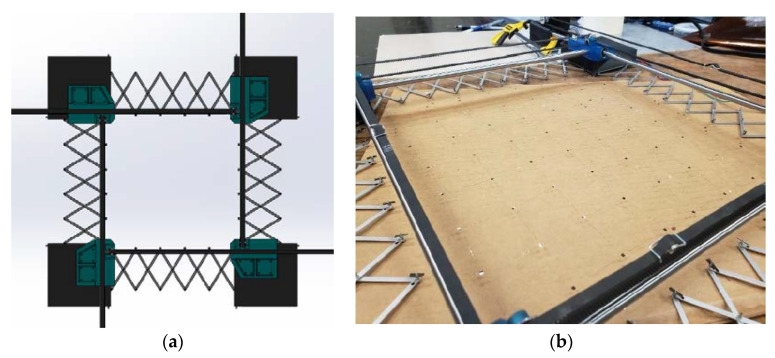
(**a**) Sensor Stretch tool design (**b**) a stretched sensor network over a large area using a sensor stretch tool.

**Figure 9 sensors-20-02528-f009:**
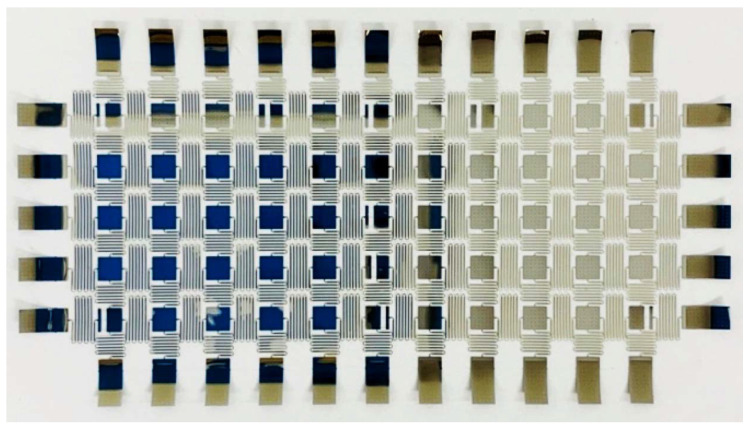
The released sensor network before stretching.

**Figure 10 sensors-20-02528-f010:**
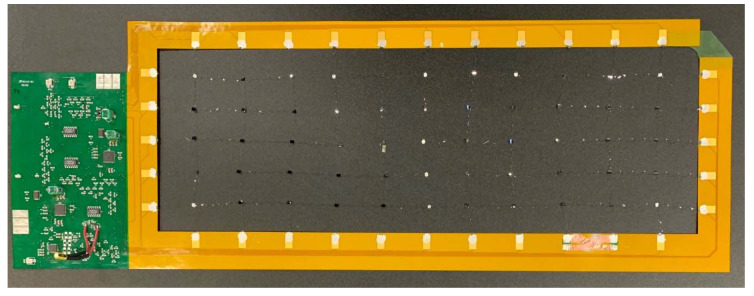
Released and stretched sensor network connected to a flexible frame with edge-mount electronics.

**Figure 11 sensors-20-02528-f011:**
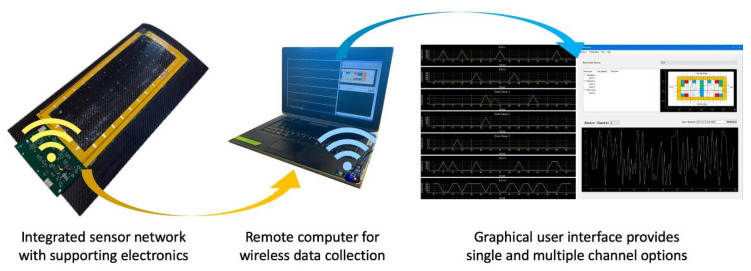
Integration of the stretched sensor network on a model aircraft wing with integrated electronic hardware and wireless communication capabilities.

**Figure 12 sensors-20-02528-f012:**
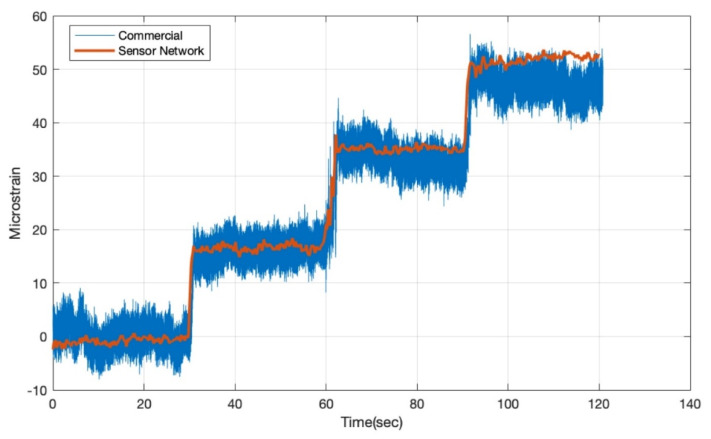
Strain measurement results of the commercial strain gauge and the strain gauge (SG1) on the sensor network near the root of the composite wing.

**Figure 13 sensors-20-02528-f013:**
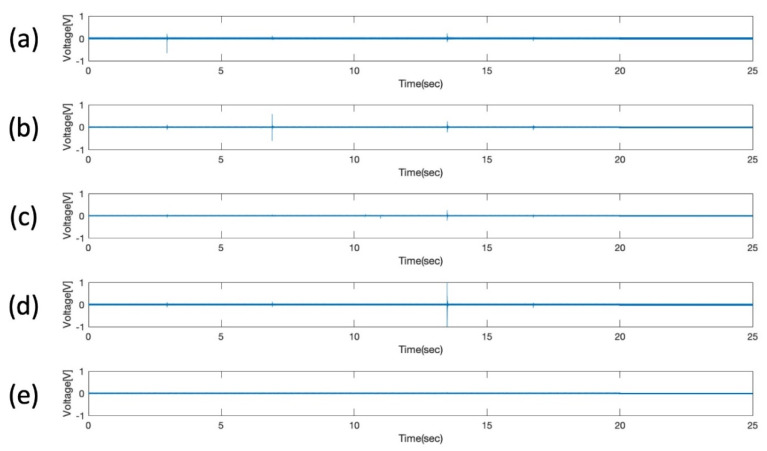
Time-domain signal from 5 PZTs at different locations on the composite wing: (**a**) PZT1; (**b**) PZT2; (**c**) PZT9 (**d**) PZT10; (**e**) PZT5.

**Figure 14 sensors-20-02528-f014:**
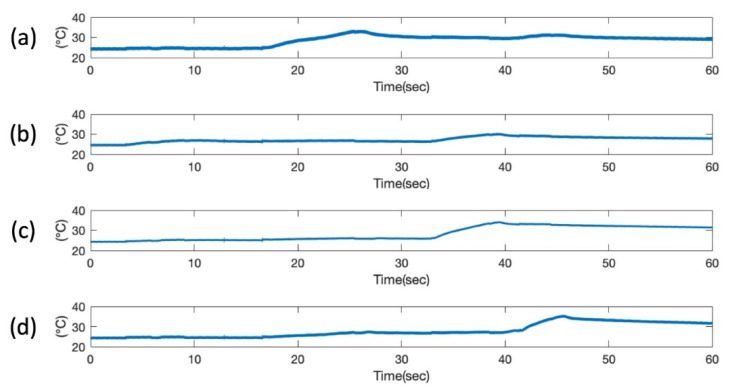
Temperature measurement results from 4 RTDs at 4 corners on the composite wing: (**a**) RTD1; (**b**) RTD2; (**c**) RTD3; (**d**) RTD4.

**Table 1 sensors-20-02528-t001:** RTD and SG simulation results in terms of interconnecting resistances.

Interconnect Resistance	Temperature Sensor Offset	Strain Gauge Resolution
300 Ω	12.3 K	0.0011 µε
205 Ω	8.5 K	0.0010 µε
70 Ω	3.0 K	0.0009 µε

**Table 2 sensors-20-02528-t002:** Averaged strain values calculated from measurement results of commercial and network strain gauges.

Time	0–30 s	30–60 s	60–90 s	90–120 s
Condition	No Loading	0.5 kg	1.0 kg	1.5 kg
Average strain measured by commercial SG	−0.1207 με	16.3715 με	33.4935 με	47.4027 με
Average strain measured by the network SG	0.7097 με	16.7150 με	35.1504 με	51.9033 με

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
