# Peer review of "Design and Integration of a Wireless Stretchable Multimodal Sensor Network in a Composite Wing†"

_sensors, 2020, doi:10.3390/s20092528_

Round 1
Reviewer 1 Report
The paper presents a stretchable sensor network which is an improvement over previously reported work. Although there are new fabrication methods presented with software development and demonstration of high SNR, the justification of problem statement in the present study seems insufficient. Following are some opportunities to make the paper more interesting.
It is straight forward that increasing the conductor area reduces resistance. Its consequence on the thickness of the sensor network and its appearance on the surface with disturbance to the aerodynamic surface is not discussed.
Explain on the shielding of sensor network and lightning protection as the sensor and its electrical connection networks are fabricated as the outer skin.
As aircraft are expected to last several decades, the practical implementation of the stretchable network for the application of protective coating and paint job, and its life expectancy should be discussed.
Author Response
Reply to Sensors
Manuscript ID: Sensors-776411
Manuscript title: Design and Integration of a Wireless Stretchable Multimodal Sensor Network in a Composite Wing
Authors are thankful to the editor, assistant editor and the reviewers for helpful comments, suggestions, and feedback. We have improved our manuscript according to the reviewers comments and
suggestions. Our responses to the comments of Reviewer #1 point by point can be found as follows.
Author's Reply to the Review Report (Reviewer #1)
We appreciate all the review comments on this work.
- The paper presents a stretchable sensor network which is an improvement over previously reported work. Although there are new fabrication methods presented with software development and demonstration of high SNR, the justification of problem statement in the present study seems insufficient. Following are some opportunities to make the paper more interesting.
- Thank you very much for your suggestions. We have made the revisions below to the manuscript to support our problem statement.
- It is straight forward that increasing the conductor area reduces resistance. Its consequence on the thickness of the sensor network and its appearance on the surface with disturbance to the aerodynamic surface is not discussed.
- Based on your comment, we have discussed this effect in Section 3.3: Fabrication Method. We have explained the benefit of increasing the thickness of the interconnects in lines 211-213 and have discussed the attendant improvement of the mechanical strength by citing previous work in line 221-224.
- The influence of the sensor network on the aerodynamic performance of the composite wing is discussed in line 416-420. We believe that this effect is negligible.
- Explain on the shielding of sensor network and lightning protection as the sensor and its electrical connection networks are fabricated as the outer skin.
- The shielding of the sensor network was discussed in our previously published conference paper. We have included a brief description only due to limited paper length. Lightning protection is mentioned in line 217-219.
- As aircraft are expected to last several decades, the practical implementation of the stretchable network for the application of protective coating and paint job, and its life expectancy should be discussed.
- The practical implementation of the stretchable network for long term aerospace applications is discussed in Section 5.2. We have cited our previous work to emphasize the reliability of our integration method and the sensor network itself.
- We do not expect the sensor network to be applicable to a commercial airplane at this stage. Our intention is to integrate it with a model aircraft to demonstrate its full functionality. Therefore, we have changed the word “airplane” to “model aircraft” in lines 425 and 430 to avoid possible misunderstandings.

Reviewer 2 Report
Thank you authors for interesting manuscript on development and implementation on stretchable multimodal sensor network.
First suggestion what comes to my attention is the quality of the figures. This is probably because of pdf-conversion. The figures were not sharp and one could not read all the information on the images. E.g. text on figure 4 and lines on figure 13 are not readable. All figures should be in better quality.
Chapter 6 Functional Tests requires improvements. The strain gauge results were written well enough.
The PZT-sensor results does not tell well of the functionality. In this demonstration should be discussed:
- did all the sensors give results (e.g. table where PZT1-10 are presented and peak voltage from each sensors from different stimulus for the wing)
- if some PZT-sensors did not give response, explain shortly why
The RTD-measurement prove that the sensor network can detect small temperature changes (from 25°C to 35ºC). It could be discussed that what were the possible limits for the temperature range?
Overall interesting manuscript. I suggest this will be accepted after additions to the functionality test results and improved image quality.
Author Response
Reply to Sensors
Manuscript ID: Sensors-776411
Manuscript title: Design and Integration of a Wireless Stretchable Multimodal Sensor Network in a Composite Wing
Authors are thankful to the editor, assistant editor and the reviewers for helpful comments, suggestions, and feedback. We have improved our manuscript according to the reviewers comments and
suggestions. Our responses to the comments of Reviewer #2 point by point can be found as follows.
Author's Reply to the Review Report (Reviewer #2)
- Thank you authors for interesting manuscript on development and implementation on stretchable multimodal sensor network.
- We appreciate your compliments.
- First suggestion what comes to my attention is the quality of the figures. This is probably because of pdf-conversion. The figures were not sharp and one could not read all the information on the images. E.g. text on figure 4 and lines on figure 13 are not readable. All figures should be in better quality.
- Thanks for pointing this out. We will upload high resolution figures to make sure they are readable.
- Chapter 6 Functional Tests requires improvements. The strain gauge results were written well enough.
- We have improved Chapter 6: Functionality Tests. We modified the manuscript according to reviewer comments.
- The PZT-sensor results does not tell well of the functionality. In this demonstration should be discussed:
did all the sensors give results (e.g. table where PZT1-10 are presented and peak voltage from each sensors from different stimulus for the wing)
if some PZT-sensors did not give response, explain shortly why
- We explained in the PZT Result that 9 out of 10 PZT sensors were giving responses. The only one that is not functioning is due to human error. This revision can be found in line 480-483.
- We also discussed the peak voltages of PZTs and briefly explained the reason why PZT5 is not as responsive to small impacts. The revision can be found in line 489-494.
- We corrected a mistake in the Figure 13 caption in line 497.
- The RTD-measurement prove that the sensor network can detect small temperature changes (from 25°C to 35ºC). It could be discussed that what were the possible limits for the temperature range?
- We cited a reference to clarify that the RTD sensor is not limited to detecting small temperature changes (i.e. from 25°C to 35˚C) and has been proven to function up to 70˚C (line 509-511).
- We also explained the reason why the cooling process is much longer than the heating process in line 505-509.
- We made some other minor modifications in line 500-503.
- Overall interesting manuscript. I suggest this will be accepted after additions to the functionality test results and improved image quality.
- Thank you again for your positive comments.
